# The Effect of Rootstock Activity for Growth and Root System Soaking in *Trichoderma atroviride* on the Graft Success and Continued Growth of Beech (*Fagus sylvatica* L.) Plants

Sławomir Świerczyński 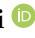





Department of Ornamental Plants, Dendrology and Pomology, Poznan University of Life Sciences, Dąbrowskiego 159, 60-594 Poznan, Poland; slawomir.swierczynski@up.poznan.pl; Tel.: +48-61-848-7955

**Abstract:** Two independent experiments were conducted on the effect of various factors, such as cultivars, growth activity of the rootstock and its treatment, with *Trichoderma atroviride* on graft success and growth of annual beech plants. The first experiment concerned the effect of propagation of five beech cultivars on rootstocks that are active (with growth activity), grown before the treatment in multi-cell plant trays (plastic seedling trays) or bare-root in the dormancy period. The highest success rate of the treatment was observed for dormant (without growth activity) and bare-root rootstocks. However, the best plant growth parameters during the first year of cultivation were observed when grafting active rootstocks obtained from multi-cell plant trays, while the worst results were observed for dormant, bare-root rootstocks. The individual cultivars varied significantly in terms of graft success and continued plant growth. The second experiment concerned the effect of rootstock growth activity and soaking of the rootstock root system in *Trichoderma atroviride* on graft success, growth parameters, and the intensity of some physiological processes in beech plants. The simultaneous use of both above-mentioned treatments resulted in the most intensive growth and accelerated physiological processes of the plants tested. Inoculation of rootstocks with *Trichoderma atroviride* did not affect the graft success. On the other hand, the growth activity of rootstocks at the time of grafting increased the success rate of the treatment. The treatments similarly differentiated the results obtained for two beech cultivars tested.

**Keywords:** beech; cultivars; propagation by grafting; physiological parameters

## 1. Introduction

Beech (*Fagus sylvatica* L.) is one of the main forest tree species and its cultivars are often used for creating urban green areas and planting in home gardens. Many cultivars of this species are currently known and 130 of which were described by Hatch [1]. Each of these cultivars has individual decorative values such as crown habit, leaf color and shape. However, it is very difficult to propagate these cultivars, and on a larger scale it is only possible by means of grafting. The success of the method of propagating ornamentals depends on many factors. The grafting technique is a very important factor. In terms of deciduous species, the most common grafting techniques include whip and tongue grafting, side grafting, wedge grafting, cleft grafting, and veneer grafting. Carey et al. [2] proved that side veneer grafts are four times more effective than top cleft grafts for grafting beech. Strong shoots from annual or biennial wood should be used for propagation by grafting ornamentals [3–6]. The physiological state of these shoots at the time of their collection is also relevant [7]; they should be harvested during exogenous dormancy [4]. The quality of the shoots collected for grafting is determined by their health status, the site where they are collected on the parent plant [8] and age [9,10]. The storage duration of scions is also important [11,12] and determines the graft success and continued growth of plants. Another factor that is closely related to the graft success is the number of buds on the scion and the length of the connection point between the rootstock and the scion, as well as the

difference in diameter between these two components [13]. The diversity of results due to the diameter of rootstocks results from the difficulty of grafting shoots that are too thin and the problems with setting the difference between the vascular cambium of the rootstock and that of the scion. It was found that graft success increases with the diameter of the rootstock [14]. In addition, the date of this treatment has a significant effect on its success rate [7]. The most common period for deciduous tree species to be grafted is from January to the end of March [3,6,15]. The graft success can be increased by earlier grafting, as it is then easier to maintain a lower temperature (10–12 °C) under the greenhouse cover or foil tunnel [13]. According to Carey et al. [2], the best time to graft beech is at the turn of winter and spring, and this is a short period of only 2 weeks; however, it can be extended (December–March) by hot callusing of the grafting site. In terms of the winter grafting of many ornamental tree species, it is advised to slow down the bud break on the scion by storing the plants for 3–6 weeks at 13–18 °C immediately after grafting, until the buds have started to grow naturally [16]. Therefore, the course of weather conditions, especially in the first few weeks after grafting, is a very important factor that affects the number of accepted grafts and their further growth [17]. An equally important determinant of graft success is the genetic, anatomical, and taxonomic consistency of the combined scion and rootstock components [18]. Some authors [6,19] believe that it is not necessary to stimulate the rootstocks for vital functions before grafting. Moreover, Dirr and Heuser [20] state that beech propagation by grafting is very difficult, as there is only a 25–30% graft success rate. Relevant experiments were conducted to test both the above-mentioned statements.

*Trichoderma* spp. is one of the most studied microorganisms due to its various mechanisms of influence on host plant growth [21]. These fungi stimulate the biosynthesis of phytohormones and other plant growth substances [22,23]. According to Benitez [24], *Trichoderma* spp. grow rapidly in a new environment and are also resistant to many toxic compounds such as fungicides, pesticides, herbicides and phenols. *Trichoderma* is one genus of saprophytic fungi that attacks and delays pathogenic fungi which cause plant diseases [25–27]. It is used for the plant (host) by foliar and soil application. However, soil inoculation is more useful because it additionally protects plants from pathogenic microbes [28]. Some researchers [24,29] state that the use of *Trichoderma* intensifies plant growth which is caused by better nutrient uptake. These fungi improve phosphorus and micronutrient uptake by releasing siderophores and secondary metabolites [30,31]. They can also produce chelating compounds that facilitate iron uptake [32]. There was a higher availability of micronutrients after T. harzianum T-203 was applied in cucumber cultivation, which increased both root and aboveground plant weight [33]. Mineral nutrition plays a key role in the vital functions of plants. Lower plant nutrition reduces net photosynthesis due to closure of the stomata [34]. There was an increase in some plant physiological parameters, such as net photosynthesis rate, stomatal conductance, and transpiration rate with improved plant nutrition [35,36]. Some *Trichoderma* strains under stress conditions can improve the photosynthesis rate and plant respiration rate by reprogramming of gene expression [37].

The effect of the state of vital functions of rootstocks at the time of grafting and of application of root system inoculation with *Trichoderma* on the success of grafting and continued growth of plants, including beech, was not previously investigated, which is what was assessed in this experiment.

## 2. Material and Methods

### 2.1. Plant Material and Growth Conditions

The first experiment propagated five beech cultivars by grafting, such as Dawyck Gold, Dawyck Purple, Purpurea Pendula, Purpurea Tricolor and Rohan Obelisk, on three groups of annual rootstocks of *Fagus sylvatica* species. This represented a total of 15 combinations of 30 rootstocks each, in triplicate. The first group of rootstocks were grown in multi-cell plant trays and stimulated to grow for two weeks before grafting at 18 °C. The second group of the rootstocks were in multi-cell plant trays, and the third group were not previously

grown in containers and represented bare-root plants. The latter two groups of rootstocks were not activated for growth immediately before grafting. In the second experiment, only two beech cultivars—Dawyck and Dawyck Gold—were grafted. Subsequently, the rootstocks were divided into four groups. Two of these groups were stimulated to grow at 18 °C, one of which was additionally treated with *Trichoderma atroviride* immediately before planting the grafted rootstocks in containers. Two further groups were not stimulated to grow, one of which was also inoculated with the above-mentioned mycelium. For this purpose, the root systems of the rootstocks were soaked in a solution of *Trichoderma atroviride* ($5 \times 10^8$ spores in g) for ease of application, and solidified with hydrogel. This experiment consisted of 8 combinations of 30 plants each, in triplicate. In both experiments, the grafting rootstocks were annual plants obtained from seed sowing. Rootstocks with a similar root collar diameter were selected for grafting. Scions of all tested cultivars came from healthy parent plants and were collected in mid-February and stored at +2 °C. They were protected from drying out until the day of grafting. The grafting was conducted in both experiments at the end of March, using side grafting. In a further step of the grafting, scions and rootstocks were tied together with flexiband and the scions were protected using Rebwachs, a wax preparation. Immediately after the treatment, the grafted plants were planted in 2 L pots in a substrate consisting of a mixture of high moor deacidified peat and finely ground pine bark in a 2:1 ratio. The substrate was further enriched with $3 \text{ g·L}^{-1}$ Osmocote Standard slow-release fertilizer. The substrate characteristics are shown in Table 1.

**Table 1.** The substrate parameter for potting cultivation of beech ($mg \cdot dm^{-3}$).

| N-NO$_3$ | P | K | Ca | Mg | pH in H$_2$O | Salnity NaCl g·dm$^{-3}$ |
|---|---|---|---|---|---|---|
| 54 | 62 | 35 | 1923 | 139 | 6.5 | 1.3 |

Prior to being grafted, the plants were grown in containers and stored indoors at 18 °C for a fortnight. In mid-April, the plants were placed in an unheated plastic tunnel. Plants were watered through micro-sprinklers. The temperature inside the tunnel with plants during the growing season was 24–28 °C by closing the diaphragm screens and opening the side vents on hot days. Preventive protection against fungal diseases was systematically performed using preparations such as Aliette 80WG (Bayer, Munich, Germany), Previcur Energy 840SL (Bayer, Munich, Germany), and Rovral Flo 255SC (BASF Agro, Ludwigshafen, Germany).

## 2.2. Morphological Plants Growth and Gas Exchange Measurements

After the end of plant growth, the percentage of graft success was evaluated, i.e., the number of plants obtained in relation to the number of rootstocks grafted. In autumn (November), measurements were taken on all the plants under study. The following observations were made: trunk diameter measured directly above the grafting site (mm), number and sum of side-shoot lengths (cm). In the first experiment, the fresh weight of leaves picked from one plant (g) was measured in triplicate. In the second experiment, physiological processes were measured during plant growth at the end of July. The measurements were taken on three plants in each of the eight plant groups. The following parameters were measured net photosynthetic rate—Pn ($\mu mol\ CO_2 \cdot m^{-2} \cdot s^{-1}$), transpiration rate—E ($\mu mol\ H_2O \cdot m^{-2} \cdot s^{-1}$), stomatal conductance—C ($mol\ H_2O \cdot m^{-2} \cdot s^{-1}$), and intracellular $CO_2$—I ($mol\ CO_2 \cdot mol^{-1}$). The measurements were performed using a manual device for photosynthesis CI-340 aa (CID Bio-Science Inc., Camas, WA, USA) and it was performed at a set intensity of active photosynthetic radiation (PAR) (1000 $\mu mol \cdot m^{-2} \cdot s^{-1}$) and a constant level of carbon dioxide (390 $\mu mol\ CO_2 \cdot mol^{-1}$ of air).

## 2.3. Data Analysis

Data analysis was processed with the STATISTICA 13.1 software (Statsoft Polska, Kraków, Poland). The obtained results were statistically analyzed using Statistica v. 12.1. A

two-way analysis of variance (cultivar, rootstock type) and a three-way analysis of variance (cultivar, rootstock activity, application of *Trichoderma atroviride*) were used in the first experiment and the second experiment, respectively, using Duncan's test at the significance level of $p = 0.05$.

## 3. Results

In the first experiment, the six beech cultivars varied in their graft success rates. The highest percentage of obtained plants was for Purpurea Tricolor and Dawyck Purple, while the lowest was for Rohan Obelisk. The way the rootstock was prepared before grafting also had a significant effect on the final graft success. The best results were observed for rootstocks that were bare-root at the time of grafting and were not activated for growth. The lowest number of plants was observed for dormant rootstocks before grafting and grown in multi-cell plant trays (Table 2).

**Table 2.** Graft success of five beech cultivars depend on type of rootstock (%).

| Cultivar | Rootstock with Active Growth | Rootstock without Active Growth | No Active Rootstock with Bare Root System | Average for Cultivar |
|---|---|---|---|---|
| 'Dawyck Gold' | 90.0 hi | 60.0 bc | 70.0 de | 73.3 b |
| 'Dawyck Purple' | 80.0 fg | 80.0 fg | 91.7 hi | 83.9 c |
| 'Purpurea Pendula' | 75.0 ef | 55.0 b | 85.0 gh | 71.7 ab |
| 'Purpurea Tricolor' | 85.0 gh | 65.0 cd | 95.0 i | 81.7 c |
| 'Rohan Obelisk' | 70.0 de | 45.0 a | 90.0 hi | 68.3 a |
| Average for rootstock | 80.0 b | 61.0 a | 86.3 c | |

Data followed by the same letters do not differ significantly at $p = 0.05$ according to Duncan's test.

Rohan Obelisk was the tallest cultivar, while Purpurea Pendula and Purpurea Tricolor were the smallest ones (Table 3). The trunk diameter of beech cultivars such as Dawyck Gold, Dawyck Purple, and Rohan Obelisk were the largest, while that of Purpurea Tricolor was the smallest (Table 4).

**Table 3.** Height of five beech cultivars depend on type of rootstock (cm).

| Cultivar | Rootstock with Active Growth | Rootstock without Active Growth | No Active Rootstock with Bare Root System | Average Value for Cultivar |
|---|---|---|---|---|
| 'Dawyck Gold' | 42.4 d | 48.4 e | 25.8 a | 38.9 b |
| 'Dawyck Purple' | 53.7 f | 42.5 d | 28.9 a | 41.7 c |
| 'Purpurea Pendula' | 40.7 cd | 27.3 a | 35.4 b | 34.5 a |
| 'Purpurea Tricolor' | 42.4 d | 36.2 bc | 26.0 a | 34.9 a |
| 'Rohan Obelisk' | 43.1 d | 48.4 e | 41.8 d | 44.4 d |
| Average for rootstock | 44.5 c | 40.6 b | 31.6 a | |

Data followed by the same letters do not differ significantly at $p = 0.05$ according to Duncan's test.

Rohan Obelisk had the highest number of side shoots, while Dawyck Gold and Purpurea Pendula had the lowest (Table 5). The sum of side-shoot lengths of Dawyck Purple were the highest, while that of Purpurea Tricolor was the lowest (Table 6). The highest fresh leaf weight was found for Rochan Obelisk, while the lowest was for Purpurea Tricolor (Table 7).

The results of all the tested beech plant growth parameters depended on the cultivation method and preparation of the rootstock for grafting. The best growth results of tested beech cultivars were found when they were grafted on active rootstocks and were previously

grown in multi-cell plant trays, while the worst results were observed for dormant and bare-root rootstocks (Tables 2–6).

**Table 4.** Diameter of stem of five beech cultivars depend on type of rootstock (mm).

| Cultivar | Rootstock with Active Growth | Rootstock without Active Growth | No Active Rootstock with Bare Root System | Average Value for Cultivar |
|---|---|---|---|---|
| 'Dawyck Gold' | 7.3 ef | 6.9 e | 5.2 a–c | 6.5 c |
| 'Dawyck Purple' | 7.7 f | 6.7 de | 5.6 bc | 6.6 c |
| 'Purpurea Pendula' | 5.9 cd | 5.6 bc | 5.5 bc | 5.7 b |
| 'Purpurea Tricolor' | 5.6 bc | 4.4 a | 5.0 ab | 5.0 a |
| 'Rohan Obelisk' | 7.3 ef | 6.6 de | 6.0 cd | 6.6 c |
| Average for rootstock | 6.8 c | 6.0 b | 5.5 a | |

Data followed by the same letters do not differ significantly at *p* = 0.05 according to Duncan's test.

**Table 5.** Number of side shoots of five beech cultivars depend on type of rootstock.

| Cultivar | Rootstock with Active Growth | Rootstock without Active Growth | No Active Rootstock with Bare Root System | Average Value for Cultivar |
|---|---|---|---|---|
| 'Dawyck Gold' | 2.2 b | 2.2 b | 1.6 a | 2.0 a |
| 'Dawyck Purple' | 3.0 e | 2.5 bcd | 1.3 a | 2.3 b |
| 'Purpurea Pendula' | 2.4 b–d | 1.4 a | 2.1 b | 2.0 a |
| 'Purpurea Tricolor' | 2.3 bc | 1.7 a | 2.3 bc | 2.1 ab |
| 'Rohan Obelisk' | 3.1 e | 2.8 de | 2.8 cde | 2.9 c |
| Average for rootstock | 2.6 b | 2.1 a | 2.0 a | |

Data followed by the same letters do not differ significantly at *p* = 0.05 according to Duncan's test.

**Table 6.** The sum of side shoots of five beech cultivars depend on type of rootstock (cm).

| Cultivar | Rootstock with Active Growth | Rootstock without Active Growth | No Active Rootstock with Bare Root System | Average for Cultivar |
|---|---|---|---|---|
| 'Dawyck Gold' | 22.8 e | 12.2 bc | 5.2 a | 13.4 ab |
| 'Dawyck Purple' | 20.2 de | 22.4 e | 12.5 bc | 18.4 c |
| 'Purpurea Pendula' | 21.1 de | 10.4 bc | 12.0 bc | 14.5 b |
| 'Purpurea Tricolor' | 13.2 c | 13.3 c | 9.0 b | 11.8 a |
| 'Rohan Obelisk' | 18.2 d | 11.5 bc | 9.1 b | 12.9 ab |
| Average for type of rootstock | 19.1 c | 14.0 b | 9.0 a | |

Data followed by the same letters do not differ significantly at *p* = 0.05 according to Duncan's test.

In the second experiment, the best graft success for both beech cultivars was achieved when the rootstocks were stimulated to grow and *Trichoderma atroviride* was not applied, while the worst graft success was observed when these two treatments were not performed (Figure 1).

The best trunk diameter was that of the tested beech cultivars where the rootstocks were active and treated with mycelium, and only one of the treatments was applied; however, it did not differ significantly (Figure 2). The smallest diameter was found in beech plants without these treatments.

The length of lateral shoots and their number for Dawyck cultivars were highest when two treatments were performed simultaneously before grafting. For Dawyck Gold cultivars, the length of lateral shoots and their number were also highest when the rootstock

was activated and *T. atroviride* was not applied. On the other hand, the absence of these treatments resulted in the lowest values of the parameters in question (Figures 3 and 4).

**Table 7.** Fresh weight of leaves of five beech cultivars depend on type of rootstock (g).

| Cultivar | Rootstock with Active Growth | Rootstock without Active Growth | No Active Rootstock with Bare Root System | Average for Cultivar |
|---|---|---|---|---|
| 'Dawyck Gold' | 7.0 fg | 6.1 def | 6.4 efg | 6.5 c |
| 'Dawyck Purple' | 11.0 j | 8.7 hi | 4.9 cd | 8.2 d |
| 'Purpurea Pendula' | 5.6 de | 4.2 bc | 4.9 cd | 4.9 b |
| 'Purpurea Tricolor' | 3.1 ab | 3.1 ab | 2.5 a | 2.9 a |
| 'Rohan Obelisk' | 11.3 j | 9.4 i | 7.5 gh | 9.4 e |
| Average for type of rootstock | 7.6 c | 6.3 b | 5.3 a | |

Data followed by the same letters do not differ significantly at $p = 0.05$ according to Duncan's test.

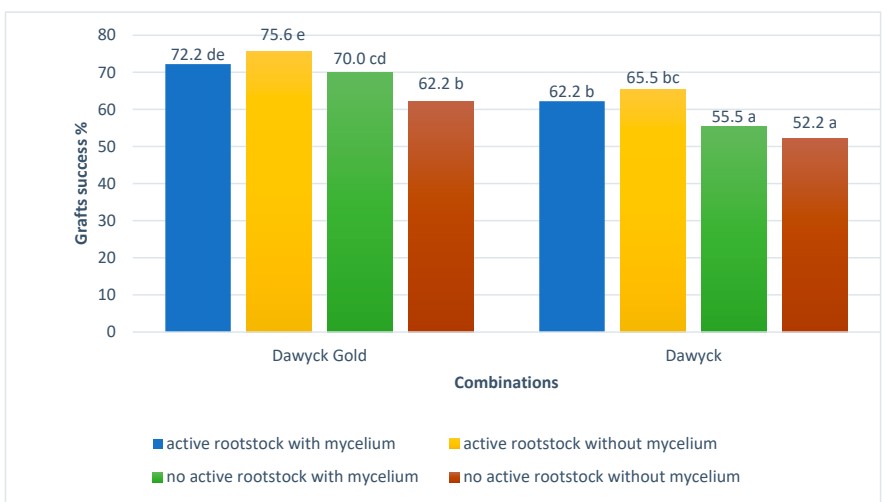

**Figure 1.** Results of the grafts succss of beech. Data followed by the same letters do not differ significantly at $p = 0.05$ according to Duncan's test.

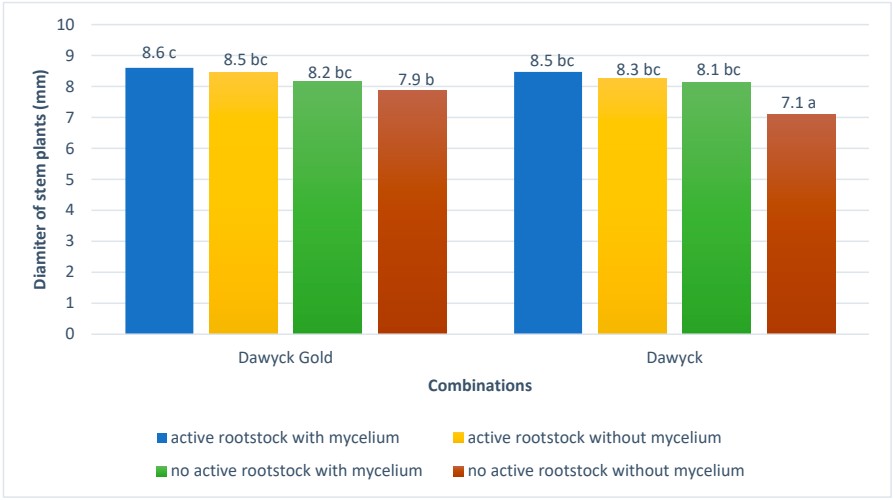

**Figure 2.** Results of the diameter of beech stem. Data followed by the same letters do not differ significantly at $p = 0.05$ according to Duncan's test.

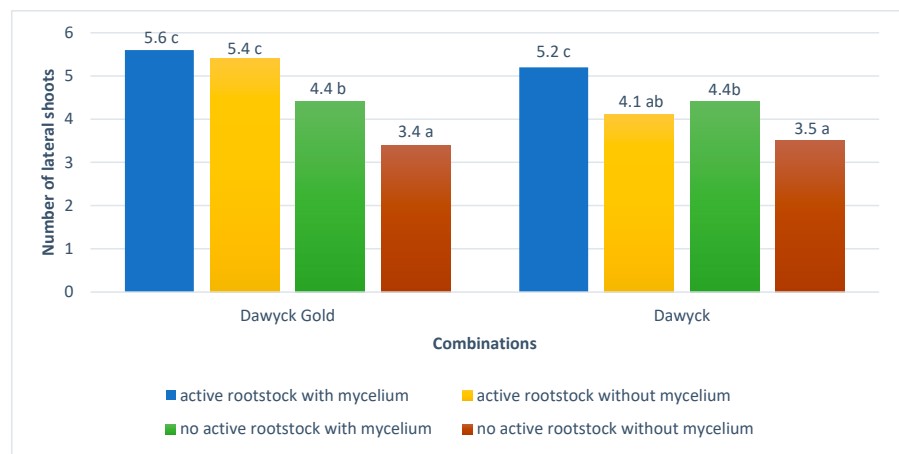

**Figure 3.** Results of the number of lateral shoots of beech. Data followed by the same letters do not differ significantly at *p* = 0.05 according to Duncan's test.

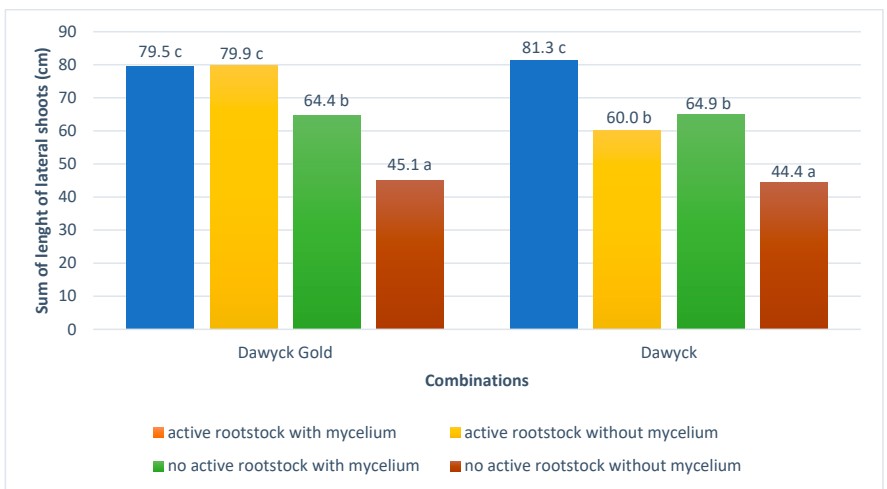

**Figure 4.** Results of sum of lenght of lateral shoots of beech. Data followed by the same letters do not differ significantly at *p* = 0.05 according to Duncan's test.

The fresh weight of beech leaves was the highest for Dawyck Gold with both treatments at the same time and for Dawyck in combination with active rootstock and the absence of fungus. The leaves of beech plants without these treatments had the lowest weight (Figure 5).

A higher net leaf photosynthesis level (Pn) was found for Dawyck compared to Dawyck Gold (Figure 6). Dawyck Gold had the best Pn index when both treatments were applied simultaneously, while Dawyck had the best Pn index only when the rootstock was treated with *Trichoderma atroviride*. The lowest value of this parameter was found for both cultivars without these two treatments.

The highest stomatal conductance index (C) for Dawyck Gold was found for two combinations where the rootstock was accelerated, while that for Dawyck was observed when both treatments were applied at the same time (Figure 7). For both cultivars, the lowest level of C index was found when no treatment was applied. The cultivar did not affect the variation of this parameter except in the control combination. The best leaf transpiration coefficient (E) for both tested cultivars was found when two treatments were applied before grafting the rootstocks, while the worst one was when no treatment was applied (Figure 8). Usually, Dawyck had higher E compared to Dawyck Gold. The internal concentration of carbon dioxide ($I\_CO_2$) in Dawyck Gold plants revealed the best results for both groups of active rootstocks. However, $I\_CO_2$ in the second cultivar revealed the best results when both treatments were applied at the same time (Figure 9). Dawyck had

higher I_$CO_2$ compared to the other cultivar tested but only when both treatments were applied at the same time.

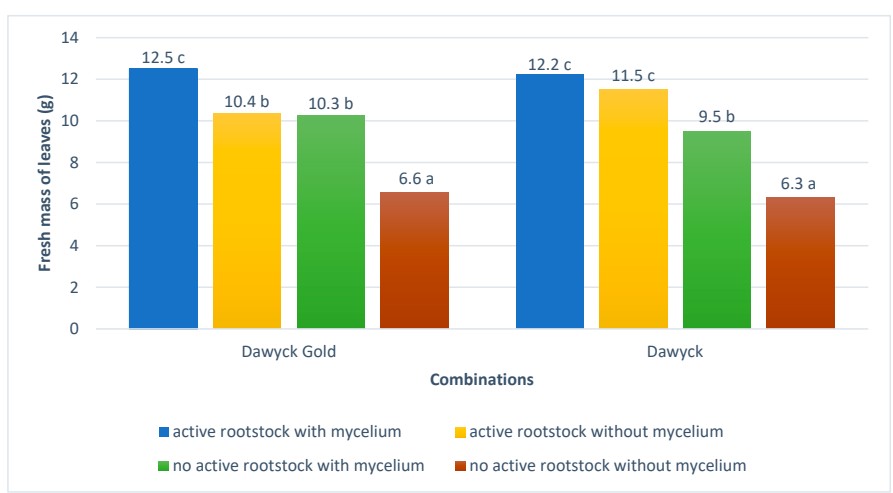

**Figure 5.** Results of fresh mass of leaves of beech. Data followed by the same letters do not differ significantly at $p = 0.05$ according to Duncan's test.

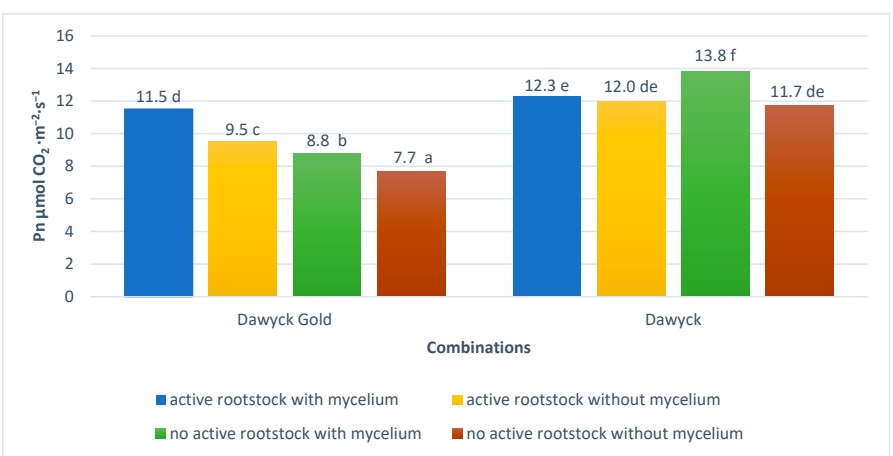

**Figure 6.** Results of Pn parameter of beech. Data followed by the same letters do not differ significantly at $p = 0.05$ according to Duncan's test.

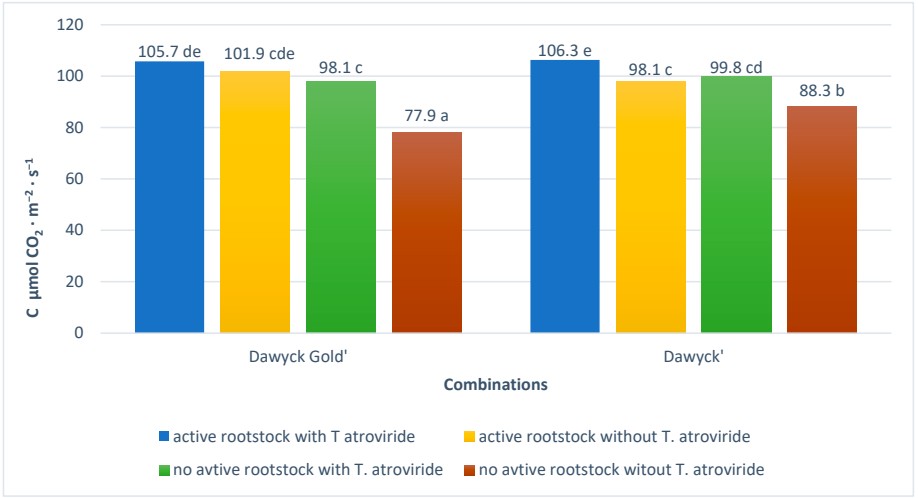

**Figure 7.** Results of C parameter of beech leaves. Data followed by the same letters do not differ significantly at $p = 0.05$ according to Duncan's test.

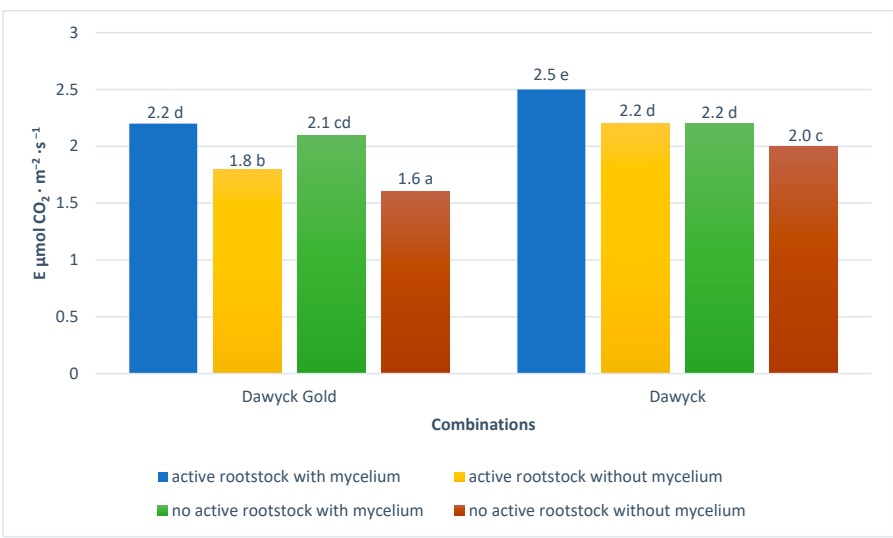

**Figure 8.** Results of E parameter of beech leaves. Data followed by the same letters do not differ significantly at *p* = 0.05 according to Duncan's test.

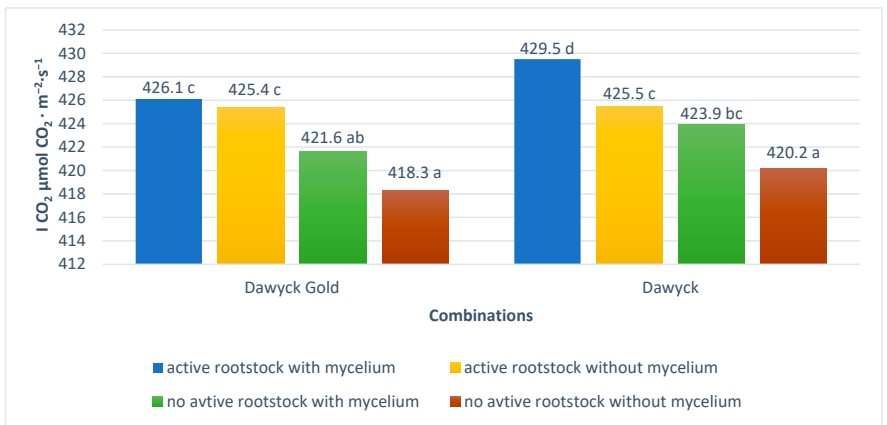

**Figure 9.** Results of I $CO_2$ parameter of beech leaves. Data followed by the same letters do not differ significantly at *p* = 0.05 according to Duncan's test.

## 4. Discussion

### 4.1. Results of First Experiment

Many studies concerning propagation by grafting of conifers [38–40] and deciduous trees [2,13,19,41] revealed that an adequately prepared rootstock of good quality and the cultivar used for grafting have a significant effect on the graft success and the growth strength of the plants obtained. Similarly as in the present experiment, the beech graft success was strongly affected by the cultivar propagated. Having propagated several beech cultivars, Nonić et al. [42] observed very different graft success (16.6–60.0%). Purpurea Tricolor had a graft success rate of 20% in their experiment, while in the present experiment it was as high as 81.7%. According to Nonić et al., this may have been a sign of physiological incompatibility between rootstocks and cultivars, but this was not proved in the present experiment. In addition, Carey et al. [2] obtained variable graft success of different beech genotypes, which is only 13% on average. In their experiment, only the use of additional hot callusing of the graft site significantly improved the success of the treatment to an average of 67%. This result was similar to the lowest one observed in the present experiment for Rohan Obelisk (68.3%) in the absence of hot callusing. The authors reported that this was due to plant health problems. The results in their experiments were also affected by technical problems and the type of grafting used. In the experiment by Ramirez et al. [13], graft success also depended on a genotype of beech propagated and it varied over the

two years of the study and was low, i.e., from 12 to 30%. A graft success rate (70%) that is similar to that in the experiment in question was found by the above-mentioned authors only when using scions with a larger diameter than the rootstock. In other combinations, with a smaller scion diameter compared to the rootstock, their graft success did not exceed 20%. Similarly, Nonić et al. [41] observed differentiated graft success of six different beech clones, which ranged from 0 to 80% and depended, among other things, on the age of grafted rootstock and grafting used. In the present experiment, the graft success rate ranged from 45.0 to 91.7% and depended on both the cultivar propagated and the rootstock growth stimulation applied before grafting. All the discussed experimental results prove the variability of beech graft success that depends on the cultivar propagated and many other factors that affect this treatment.

In the conducted experiments, there was a constant temperature of approx. 18 °C in the room where plants were kept immediately after they were grafted, which resulted in a high percentage of graft success for several beech varieties. This temperature was also recommended by Dirr and Heuser [20] in the first three weeks after grafting. In the present experiment, there were significantly more cases of graft acceptance compared to those suggested by the aforementioned authors (25–30%). They claim that a good result of grafting is achieved when the scion has a similar diameter to the rootstock. A similar rule was followed in the present experiment, which had a positive effect on the grafting results obtained. A similar graft success was not obtained for all tested beech cultivars, which could be due to the variable diameter of scions taken from different parent plants. However, a similar scion and rootstock diameter was used for each cultivar. Rootstock quality is a critical factor that affects graft success [43]. It is believed that the rootstocks should be disease-free [20], which was the case in the experiments conducted. Some authors [2,34] state that hot callusing at scion-rootstock interface should be recommended in the case of deciduous species with low graft success. However, our own observations do not prove the necessity of using this treatment to obtain a good result in beech grafting. When compared with the graft success results of Carey et al. [2], which were obtained by applying hot callusing, there were similar results in the present experiment without applying this treatment. In the present experiment, excluding the graft success, there was a positive effect of the rootstock activity at the time of grafting for other analyzed plant growth parameters. It is not in line with the opinion of other authors [4,6,19] who indicate that such rootstock activity is not necessary for beech grafting compared to other tree species, e.g., oak. The strength of the growth of beech plants depended on the nature of the growth of the cultivar propagated. Strong-growing cultivars such as Rohan Obelisk and Dawyck Purple had higher growth parameters. On the other hand, the low-growing cultivars such as Purpurea Pendula and Dawyck Gold had lower growth that was determined by the same parameters. Dawyck Purple stands out in terms of the number and length of side shoots. This also highlights the nature of its growth. Dawyck Purple forms a compact crown with a large number of side shoots. In the second experiment, strong-growing Dawyck had slightly better growth parameters compared to Dawyck Gold which grows more slowly. However, the plant growth of these cultivars was also affected by two additional treatments that were applied. The correct difference in terms of growth of tested cultivars may also only become apparent in subsequent years of cultivation. The results of trunk diameter in both experiments depended on the cultivar and were significantly worse compared to those obtained by Nonić et al. [41]. However, it is difficult to compare the results of both experiments because the authors did not specify the height at which the rootstock was grafted, and the trunk diameter measured 30 days after grafting was much larger than that in the present experiment, which was obtained after the first year of the growth of beech plants. Therefore, it can be concluded that the above-mentioned authors used rootstocks with a larger diameter.

*4.2. Results of Second Experiment*

In the second experiment, in addition to the use of active rootstocks, a simultaneous inoculation of their root system with a fungus was applied. A positive effect of this treatment was achieved for both active and dormant rootstocks. Similarly, other authors found [44] a 42–44% improvement in the growth of mango plants that were propagated by grafting. The growth was expressed by shoot length after the use of *Trichoderma* sp. by wetting the cutting surface of scions during grafting. In the present experiment, there was a very similar increase in side shoot growth which was 43–46% according to the cultivar grafted. In another experiment [45], when propagating by grafting cocoa plants, *Trichodema asperellum* was used at the cutting site of the rootstock. After inoculation, there was an increase of 90% in the number of side shoots. According to the above-mentioned authors and others [25,46], such a large improvement in growth may be due to the fact that *Trichoderma* affects the plant to produce more auxins and fewer cytokinins and abscisic acid, which results in stronger plant growth. The positive effect of *Trichoderma* on reducing disease incidence and, consequently, better plant growth was proved [47,48]. The same situation could occur in the present experiment, where better growth results of beech plants were obtained after *Trichoderma atroviride* was applied. According to some researchers, improved plant growth is due to improved nutrient uptake [24,29], including micronutrients [33]. Harman et al. [49] point to a mechanism that leads to the bioavailability of elements. The achievement of better plant growth is also due to the ability of *Trichoderma* sp. to produce secondary metabolites (peptaibols, harzianolides) and other metabolites that promote shoot growth [50,51]. However, Di Vaio et al. [28] did not obtain any improvement in growth parameters of young olive trees in a nursery after the commercial preparation Trianum-P (*Trichoderma harzianum*) was applied. In their experiment, only the total leaf area was 11% larger compared to controls. In our experiment, fresh leaf weight was as much as 50–56% higher. This relationship is confirmed in the studies by Yedidia et al. [33] on cucumbers, and by Vinale et al. [52] and Azarmi et al. [53] on tomatoes. In addition, Rakibuzzaman et al. [54] state that *Trichoderma* improves graft success and plant growth, which was observed in the experiment in question. These authors observed an improvement in the photosynthetic rate in tomatoes after *Trichoderma* was applied [54]. In the experiment by Rosman et al. [45], the application of *T. asperellum* during cocoa grafting also increased plant photosynthesis, mainly through higher $CO_2$ uptake. Rosman et al. also observed a greater opening of the stomata and their higher number, which was correlated with stronger seedling growth. Both our own observations and those of all the above-mentioned authors confirmed the acceleration of physiological processes in plants after *Trichoderma* was applied.

**5. Conclusions**

Both experiments in question revealed improved growth of beech plants when propagated with grafting with the use of an active rootstock and its additional inoculation with *Trichoderma atroviride*. The graft success of beech depended less on the rootstock activity at the time of grafting and hardly at all on the use of rootstock inoculation with *Trichoderma atroviride*. In the first experiment, it was found that a better a graft success rate was observed for dormant and bare-root rootstocks. It can be concluded that the application of additional treatments in question, which are related to the preparation of rootstocks for grafting, improves the growth of propagated beech plants but does not affect the graft success. In view of the diversity of results obtained to date, further research is needed to improve this method of deciduous tree propagation.

**Funding:** Publication was co-financed within the framework of the Polish Ministry of Science and Higher Education's program: 'Regional Initiative Excellence' in the year 2019–2022 (No. 005/RID/2018/19), financing amount 1,200,000 PLN.

**Institutional Review Board Statement:** Not applicable.

**Informed Consent Statement:** Not applicable.

**Data Availability Statement:** Not applicable.

**Conflicts of Interest:** The authors declare no conflict of interest.

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
