# Peer review of "The Effect of Rootstock Activity for Growth and Root System Soaking in Trichoderma atroviride on the Graft Success and Continued Growth of Beech (Fagus sylvatica L.) Plants"

_agronomy, doi:10.3390/agronomy12061259_

Round 1

Reviewer 1 Report

The manuscript is generally well prepared and language could be easily readable, but improvement should be given, particularly on description of experiment design, results and discussion, and careful check on grammar and spelling. The manuscript must be revised mainly on the following comments.

  1. Experimental design should be more clearly described in more details, such as when the grafting success data are collected. A table is suggested to use for experimental design, which will be clearer.
  2. The subtitles should be used for both results and discussion parts to be easily readable.
  3. In the conclusion, the author mentioned “in the first experiment”, but how about on the second experiment? It is suggested to show what experiment for both, not to say in the first or the second experiment.

Much work should be done on language editing in most marked places, mainly including:

  1. Abstract need to be more concise
  2. Some grammar errors need to be correct.
  3. Misspelling check is needed in many places, such as activ (active), five beech cultivar (cultivars) in tables.
  4. Add subtitles for a long discussion to make it more clear in presentation and logic.
  5. On citation, 15 should be listed before 16, and 54 is missed.
  6. Literature should be checked in many places, on italic form for Latin names, and low and capitalized letters for article titles, and consistent spelling form on using either full spelling or abbreviation for journal names based on journal requirement.
  7. Use double length of hyphen but not hyphen for values range.
  8. Other comments are marked in revised text.

Author Response

Thank you for all valuable comments. The shown errors have been corrected.
However, I have doubts whether it is possible to interfere with the original titles
of publications included in the list of publications, where the authors did not
always write the latin name italics.
It is also not possible to change the italics of Latin names in the description
of explanations inside the figures (no technical possibilities)
therefore the
species name was changed to mycelium.

Reviewer 2 Report

Line 111 instead „root neck” write „root collar” diameter which is a correct technical term.

Line 116 Instead of „Redwachs” probably you applied „Rebwachs”

Formatting of all Tables should be corrected fitted tot he text

Line 133 – instead of plant diameter use trunk diameter

In fig 1 to 7 decide the writing form of varieties with both apostrophe (e.g. ’Dawyck Gold’) or without (Dawick Gold). In all the Figures the first apostrophe is missing.

In Fig 3 please delete the Polish word „Odmiana” (variety), it is not necessary.

Fig 3 and 4 : please correct the word „latheral” to „lateral”. (The meaning of latheral is completely different, is not in correspondence with your intention). Write to the vertical axis of the Figures 3 and 4 „Number of laterals”

Line 343:  instead of „In the experiment in question” write „In our experiment”; so the sentence is more exact. The term „In the experiment in question” may refer to the previous referred author too.

Line 362: to be more clear, instead of „treatments in question” write „Trichoderma treatments”.

Author Response

Thank you for any valuable comments that have been taken into account.
In the conclusions only the suggested change was not introduced because this remark
does not only concern the effect of the Trichoderma fungus but also the activity of
the rootstock for growth
